# Unraveling the Phase Transition Behavior of MgMn_2_O_4_ Electrodes for Their Use in Rechargeable Magnesium Batteries

**DOI:** 10.3390/ma16155402

**Published:** 2023-08-01

**Authors:** Carmen Miralles, Teresa Lana-Villarreal, Roberto Gómez

**Affiliations:** Departament de Química Física i Institut Universitari d’Electroquímica, Universitat d’Alacant, Apartat 99, E-03080 Alicante, Spain; carmen.miralles@ua.es

**Keywords:** MgMn_2_O_4_, magnesium batteries, in-situ Raman spectroscopy, Mg^2+^ insertion

## Abstract

Rechargeable magnesium batteries are an attractive alternative to lithium batteries because of their higher safety and lower cost, being spinel-type materials promising candidates for their positive electrode. Herein, MgMn_2_O_4_ with a tetragonal structure is synthesized via a simple, low-cost Pechini methodology and tested in aqueous media. Electrochemical measurements combined with in-situ Raman spectroscopy and other ex-situ physicochemical characterization techniques show that, in aqueous media, the charge/discharge process occurs through the co-intercalation of Mg^2+^ and water molecules. A progressive structure evolution from a well-defined spinel to a birnessite-type arrangement occurs during the first cycles, provoking capacity activation. The concomitant towering morphological change induces poor cycling performance, probably due to partial delamination and loss of electrical contact between the active film and the substrate. Interestingly, both MgMn_2_O_4_ capacity retention and cyclability can be increased by doping with nickel. This work provides insights into the positive electrode processes in aqueous media, which is vital for understanding the charge storage mechanism and the correlated performance of spinel-type host materials.

## 1. Introduction

The increasing penetration of renewable energy sources and the extended use of portable devices and electric vehicles put pressure on established energy storage technologies, such as Li-ion batteries. Researchers worldwide seek alternatives to improve storage capacity, power, safety, and/or cost. In this respect, magnesium batteries have been regarded as a potentially viable solution since the late 80s. The physicochemical properties of Mg could enable batteries that outperform, in some respects, existing Li-ion batteries by offering, for instance, higher energy densities. Magnesium has a theoretical volumetric capacity higher than lithium’s (Mg: 3833 mAh cm^−3^, Li: 2062 mAh cm^−3^), which can be particularly advantageous for space-limited applications. In addition, magnesium-based batteries could be more cost-effective than lithium counterparts as magnesium is three orders of magnitude more abundant than lithium in the Earth’s crust. Additionally, magnesium batteries would be environmentally friendlier, especially if we consider the waste of the lithium metal extraction processes.

Despite these advantages and the fact that the research in magnesium batteries has been significant for about 30 years, the commercialization of magnesium metal rechargeable batteries is still far from reality. The major bottleneck to be addressed is the difficulty of combining electrolytes and electrodes, allowing for reversible Mg insertion-extraction at potentials giving rise to a reasonable energy density [1]. In general, the limited activity of electrode materials has been related to the small size and high charge density of the Mg cation, which explains its high polarizing power. This triggers the destabilization of the host cathode framework, and it reduces the Mg ion mobility. Furthermore, the high polarizing power of Mg ions induces their strong interaction with complex counterions in the electrolyte [2,3]. There are only a few cathode materials that show significant electrochemical activity toward Mg in organic electrolytes, such as V_2_O_5_ [4,5], MnO_2_ [6,7] and Mo_6_S_8_ [8]. Among them, the Chevrel phase Mo_6_S_8_ is the only one with relatively good cycling stability. However, its synthesis process is complex, and the potential window in which the redox reactions occur is at relatively low potentials, between 1.0 and 1.3 V vs. Mg^2+^/Mg [8].

The spinel MgMn_2_O_4_ has been proposed as a cathode material for magnesium batteries [9]. Although Mg^2+^ can be reversibly inserted in the MgMn_2_O_4_ structure leading to Mg_2_Mn_2_O_4_ (Mn^3+^/Mn^2+^), the reversible extraction of Mg from MgMn_2_O_4_ is more attractive. The corresponding Mn^4+^/Mn^3+^ redox process leads to a theoretical capacity of 272 mAh g^−1^ while allowing Mg^2+^ insertion/extraction to take place at potentials more positive than for other types of cathode materials such as sulfides or selenides [10,11].

Mg insertion-extraction in the MgMn_2_O_4_ lattice has been demonstrated both in organic media using Mg(ClO_4_)_2_ [12] and Mg(TFSI)_2_ as electrolytes [13] and in aqueous media with Mg(NO_3_)_2_, MgCl_2_ and MgSO_4_ salts [14,15,16]. Several reports have proven that the charge/discharge process involves a phase transition from the original spinel MgMn_2_O_4_ structure. In aqueous media, the insertion of water during electrochemical cycling has been reported with a concomitant phase transition to a laminar structure [17,18,19]. In organic media (acetonitrile) and ionic liquids (CsTFSA/Mg(TFSA)_2_, TFSA: bis(trifluoromethanesulfonyl)amide), in the absence of water, the cubic spinel structure evolves toward a tetragonal one [20,21]. Consequently, spinel MgMn_2_O_4_ often presents an initial capacity that declines quickly with the number of cycles in organic and aqueous media. The claimed reasons behind this behavior are (1) the already mentioned phase transition occurring upon charge-discharge; (2) the dissolution of Mn with the number of cycles due to the disproportionation reaction of Mn(III) favored by the Jahn Teller effect; (3) the constraints to extract Mg from the MgMn_2_O_4_ lattice due to the intrinsic slow diffusion of Mg^2+^ in solid phases [2,22]. Different strategies have been implemented to enhance and stabilize the electrochemical behavior of MgMn_2_O_4_. Most of them pursue limiting the lattice distortion by favoring a cubic structure [23], which can be promoted through the inhibition of the Jahn-Teller effect by either doping [24] or generating 3-D open-channel nanostructures [25]. Introducing rGO or carbon nanotubes also induces remarkable capacity stabilization [26,27].

In the case of Li insertion/extraction, doping with transition metals has been revealed as a promising methodology to promote stabilization by hindering the dissolution of Mn [28,29]. This approach has also been implemented for Mg insertion/extraction. Recently, a slight increase in the structural capacity retention and cyclability has been shown for MgMn_2_O_4_ spinels doped with Fe [30,31]. MgMn_2_O_4_ spinels have also been doped with cobalt (MgCo_0.5_Mn_1.5_O_4_) and nickel (MgNi_0.5_Mn_1.5_O_4_) by Banu et al., although the doped oxides did not show advantages in terms of capacity compared to the pristine MgMn_2_O_4_ spinel [32].

Little information is available in the literature about the critical importance of the spinel host structure for reversible magnesium insertion/extraction. Certainly, the initial structure configuration is crucial for electrode performance and ion transport. Fundamental research on this topic should contribute to progress in battery technology, expanding our understanding of electrochemical energy storage systems and contributing to the development of magnesium batteries. As already mentioned, these batteries offer a promising alternative that can help to overcome some of the shortcomings of lithium batteries, such as limited lithium availability. In this work, we examine the electrochemical behavior of MgMn_2_O_4_ in aqueous electrolytes. Electrochemical measurements combined with in-situ Raman spectroscopic experiments and ex-situ microscopic and spectroscopic measurements have allowed us to examine further the Mg extraction/insertion process and the oxide structural evolution. The effect of transition-metal ion substitution (cobalt and nickel) in MgMn_2_O_4_ has also been studied, concluding that doping with Ni can modify the initial spinel framework, improving capacity retention.

## 2. Materials and Methods

### 2.1. Synthesis of MgMn_2_O_4_ and MgMn_2_O_4_ Doped with Either Co or Ni

MgMn_2_O_4_ was synthesized by employing the Pechini sol-gel method [13]. Specifically, a solution containing 0.5 M Mg(NO_3_)_2_·6H_2_O (VWR chemicals BHD, 99%, Leuven, Belgium), 1.0 M Mn(NO_3_)_2_·4H_2_O (Sigma Aldrich, 97%, St. Louis, MO, USA) in 5 mL ethylene glycol (Sigma Aldrich, 99.8%) and 5 mL H_2_O was stirred for 30 min. The citric acid (Sigma Aldrich) was then added to obtain a 3.0 M solution, which was stirred for 1 h.

The solution was maintained at 70 °C for 24 h, triggering the generation of a brown gel, which was next thermally treated in an oven at 200 °C for 12 h. The resulting black solid was ground and heated at 400 °C in the air for 10 h. The MgMn_2_O_4_ spinels doped with either cobalt or nickel were synthesized in the same way but adding to the precursor solution 1.6 mmol of Co(NO_3_)_2_·6H_2_O (Sigma Aldrich, 98%) in the case of the Co-doped spinel and 1.6 mmol of Ni(NO_3_)_2_·6H_2_O (Sigma Aldrich, 97%) for the Ni-doped spinel.

### 2.2. Characterization of MgMn_2_O_4_ and MgMn_2_O_4_ Doped with Either Co or Ni

The crystal structure of MgMn_2_O_4_, both pristine and doped with Co or Ni, was identified by X-ray Diffraction (XRD) (Bruker D8-Advance instrument equipped with a Kristalloflex K 760-80F X-ray generator with a Cu anode operating at 15 kV and 10 mA). Both the electrodes and the as-synthesized powders were measured between 10 and 80° at 1 degree min^−1^. The crystal structure was also characterized with an NRS-5100 Raman microscope from Jasco with a 17 mW He-Ne laser (632.8 nm). The laser was focused through a 50 × LWD objective into a 2 µm spot at the sample surface. The XPS spectra were recorded using a Thermo-Scientific K-Alpha XPS spectrometer equipped with a monochromatic Al-K source operating at 15 kV and 10 mA and an Ar^+^ source for surface etching. The film morphology and particle size were analyzed using a ZEISS Merlin VP compact field emission scanning electron microscope (FE-SEM) and a JEOL JEM-2010 transmission electron microscope (TEM). The different synthesized materials were analyzed by inductively coupled plasma optical emission spectroscopy (ICP-OES) using Optima 7300 DV equipment from Perkin Elmer. For this purpose, the electrodes were dissolved in a 0.5 M HCl aqueous solution.

### 2.3. Electrochemical Experiments

The electrochemical measurements, cyclic voltammetry and galvanostatic charge-discharge cycles were carried out with an AUTOLAB PGSTAT30 potentiostat-galvanostat. A three-electrode glass cell was employed, equipped with a Pt wire as a counter electrode and a commercial AgCl/Ag/KCl (3.0 M) as a reference electrode to which all potentials are referred. The working electrode was prepared by using a slurry containing MgMn_2_O_4_ nanoparticles, carbon black super P (Thermo scientific, 99+%, Kandel, Germany) and polyvinylidene fluoride, PVDF (Sigma Aldrich) (80:10:10) in N-methyl-2-pyrrolidone (NMP). A 0.1 mm thickness titanium foil (99.6%, Goodfellow, Huntingdon, UK) was employed as a cathode substrate. The substrate was cut as 2 × 1 cm^2^ rectangular pieces. The slurry containing the active material, carbon black, and PVDF was deposited on an area of 1 cm^2^.

The electrodes were dried at 80 °C for 14 h under vacuum before their use, the mass of the active material per geometric area being 1.92 mg cm^−2^. All the specific values refer to the active material mass. The electrochemical behavior was studied in nitrogen-purged 0.1, 1.0 and 3.0 M Mg(NO_3_)_2_ aqueous electrolytes.

In situ Raman experiments were carried out with a three-electrode Teflon spectroelectrochemical cell equipped with a quartz window, a Pt wire as a counter electrode, and an Ag/AgCl/KCl (3.0 M) electrode as a reference electrode. For these experiments, a nitrogen-purged 1.0 M Mg(NO_3_)_2_ solution was employed, and the working electrode was prepared by depositing the MgMn_2_O_4_ slurry on a polished glassy carbon rod.

## 3. Results

### 3.1. Physicochemical Characterization of the As-Obtained MgMn_2_O_4_

The structural characterization of the as-obtained magnesium manganese oxide (MgMn_2_O_4_) powder synthesized using the Pechini method was carried out by means of X-ray diffraction and Raman spectroscopy. Figure 1a shows the diffractogram with the characteristic peaks corresponding to a MgMn_2_O_4_ spinel with a tetragonal structure. It differs from the typical cubic structure due to the distortion induced by the strong Jahn-Teller effect of Mn^3+^ [33,34,35]. The elongation of the two apical Mn-O bonds causes the crystal structure to widen along the c-axis. In agreement, the XRD pattern can be fitted to a tetragonal spinel MgMn_2_O_4_ structure with space group I41/amd, according to the 01-072-1336 JCPDS card.

Raman spectroscopy is a non-destructive powerful tool that allows the characterization of the crystal structure, even in the case of a limited range order. However, up to now, its use for MgMn_2_O_4_ characterization has been scarce. According to the group theory, ten Raman active modes (2A_1g_ + 3B_1g_ + 1B_2g_ + 4E_g_) are expected for the tetragonal spinel structure. The spectrum is characterized by a strong peak at 652 cm^−1^ and two less intense bands at 295 and 354 cm^−1^ (Figure 1b). The most intense peak at 652 cm^−1^ can be ascribed to vibrations involving the motion of oxygen atoms inside the MnO_6_ octahedral units, specifically to the A_1g_ mode (Mn-O stretching vibration).

Interestingly, such a band is sensitive to the unit cell volume and the spinel inversion degree [33]. By analogy with the LiMn_2_O_4_ Raman spectrum (Appendix A) [36,37], the less intense band located at 354 cm^−1^ can be assigned to Mg-O vibrations of MgO_4_ groups, while the peak at 295 cm^−1^ would correspond to the bending vibrations of MnO_6_ groups [36,37]. Although the XRD and Raman results are consistent with a tetrahedral crystal structure, it is essential to mention that the frequencies for MgMn_2_O_4_ reported here are shifted with respect to those reported by Flor et al. [33]. This may be due to a lack of stoichiometry in our case [38]. The Pechini method employed in this work is based on the formation of chelates involving mixed cations (Mg and Mn in solution) and a hydroxycarboxylic acid (citric acid in our case) that are cross-linked during the synthesis to create a gel through esterification. Finally, the gelled composites are thermally treated, pyrolyzing the organics and generating the final nanoparticles. In the different steps, and particularly during the thermal treatment, the initial ideal stoichiometric composition could be lost. In this regard, the electrode elemental analysis revealed, in fact, a lack of stoichiometry, with an Mg/Mn ratio of 0.40 (Mg_0.8_Mn_2_O_4±δ_). This can lead to the generation of vacancies, which, combined with a small nanoparticle size, could favor magnesium insertion/extraction. Figure 1c shows a TEM image for the generated nanoparticles. They have a polyhedral shape with an average size of 16 nm (Figure 1d).

### 3.2. Electrochemical Behavior of MgMn_2_O_4_ in an Aqueous Mg(NO_3_)_2_ Electrolyte

According to the literature, MgMn_2_O_4_ may undergo two redox reactions involving the Mn^3+^/Mn^2+^ and Mn^4+^/Mn^3+^ couples:(1)$${\mathrm{MgMn}}_{2}O_{4}+{\mathrm{Mg}}^{2+}+2e^{-}↔{\mathrm{Mg}}_{2}{\mathrm{Mn}}_{2}O_{4}$$
(2)$${\mathrm{MgMn}}_{2}O_{4}↔2{\mathrm{MnO}}_{2}+{\mathrm{Mg}}^{2+}+2e^{-}$$

Both reactions imply the insertion/extraction of Mg^2+^ ions, enabling their exploitation in batteries. The stability of the spinel framework and the electrochemical response are highly dependent on the working potential window and electrolyte composition. Takeuchi et al. showed that the water content in non-aqueous electrolytes plays a key role in increasing the reversibility of the electrochemical processes and the electrode cyclability [13]. In this context, we have studied the magnesium insertion-disinsertion from MgMn_2_O_4_ in aqueous media.

Cyclic voltammetry (CV) has been employed to select the proper electrolyte concentration. Figure 2a shows pseudo-stationary cyclic voltammograms for different Mg(NO_3_)_2_ concentrations. As concentration increases from 0.1 to 3.0 M, the voltammetric peaks become better defined, and the area under the CV curve rises. Furthermore, the difference between the main anodic and cathodic peak potentials decreases. The behavior observed for low Mg^2+^ concentrations can be ascribed to free-electrolyte starvation. In a porous 3D electrode like that employed here, almost all the ions in the pores become adsorbed/inserted at the high-area interface, enhancing the internal electrolyte resistance effect. When the concentration is 1.0 M or higher, such a limitation is no longer detected, and more defined and larger quasi-reversible voltammetric peaks are observed. Furthermore, as expected for a Nernstian behavior, the voltammetric peaks shift toward more positive potentials as magnesium concentration increases. In any case, the dependence of the cyclic voltammograms on Mg^2+^ concentration indirectly indicates the insertion-extraction process.

For the subsequent experiments, 1.0 M Mg(NO_3_)_2_ aqueous solutions were employed. Figure 2b shows the evolution of the voltammetric profile with the scan number for a MgMn_2_O_4_ electrode. In the first cycle, a growing anodic current can be observed starting from 0.2 V, which is transformed into two well-defined anodic peaks in subsequent cycles. This indicates difficulties in extracting Mg^2+^ from pristine MgMn_2_O_4_. After five cycles, two oxidation peaks and one broad reduction peak containing different contributions are defined. In addition, an increase in the current magnitude gradually occurs upon cycling. The fact that the anodic and cathodic charge ratio approaches one points to the fact that the CV peaks correspond to the insertion-extraction of Mg^2+^ into/from the MgMn_2_O_4_ lattice. The observed capacity activation suggests a structure evolution process in the initial cycles, which, unfortunately, leads to a deterioration of the electrochemical behavior. Figure 2b shows that beyond 33 cycles, currents significantly diminish. Such behavior is also reflected in the charge/discharge profiles at various current rates (Figure 3). The curves, including their definition, evolve with the number of cycles. For cycle number 5, at 0.5 mA (250 mA g^−1^) and 0.1 mA (53 mA g^−1^), two regions are observed at approximately 0.2 V and 0.03 V in the discharge curve, corresponding to two Faradaic reduction processes. In the load curve, the presence of both steps is also observed but in a less defined way. At 0.1 mA, the definition of the processes is higher than at 0.5 mA, as expected. In any case, the charge-discharge curves are fully compatible with the CVs in Figure 2.

The capacity progressively decays with the cycle number. At 0.5 mA, the maximum capacity value for the spinel during discharge is 172 mAh g^−1^, with a 64% capacity retention in cycle 45 (Figure 3b). At 0.1 mA, the maximum capacity is 182 mAh g^−1,^ and the capacity retention in cycle 45 is 53% (Figure 3d). The capacity loss is more pronounced at lower currents, consistent with a Coulombic efficiency at 0.5 mA (98.7%) higher than at 0.1 mA (84%). At lower current densities, the slower rate of potential change promotes side reactions close to the limits of the potential window. The main side reaction would be the Mn dissolution at low potentials during the discharge (reduction) because of extended Mn(II) formation.

On the other hand, the side reaction at the end of the charging process would be an incipient oxygen evolution, resulting in larger apparent capacities during the charge. The latter is the main process explaining the low Coulombic efficiency al low charge rates. The capacity and cycling performance of the as-prepared MgMn_2_O_4_ cathode are among the best reported in aqueous media for spinel- and birnessite-type magnesium host materials [15,26,31].

### 3.3. Electrode/Electrolyte Interfacial Reaction during Cycling

Revealing the nature of the electrode processes is vital for understanding the charge storage mechanism and the correlated performance of MgMn_2_O_4_. X-ray photoelectron spectroscopy has been used to analyze the chemical composition and oxidation state at the surface during the electrochemical processes. Specifically, the Mg 1s and Mn 2p transitions have been analyzed when the electrode is reduced at −0.7 V and oxidized at 1.0 V, and for the pristine material (freshly prepared). Table 1 gathers each case’s Mg/Mn ratio, including surface and in-depth analysis after a 20 nm etching treatment by bombardment with Ar^+^ ions. The results show that the as-synthesized spinel has a significant Mg surface deficiency, which agrees with the elemental analysis of the sample by ICP-OES (Mg_0.85_Mn_2_O_4±δ_). More importantly, these data prove that magnesium is inserted into the spinel structure: the Mg/Mn ratio is much higher when the spinel is reduced. The fact that the values for the etched sample show the same trend proves that insertion is not exclusively limited to the outer part of the film. However, comparing the values before/after etching (surface/bulk content) for the oxidized and reduced samples reveals opposite trends. The ratio Mg/Mn is lower upon etching for the reduced electrode, while it is higher for the oxidized electrode. This can be rationalized based on the slow diffusion kinetics of Mg in the host spinel. When magnesium is inserted, it does so to a larger extent in the outer part of the film. On the other hand, for the oxidized sample, the Mg/Mn ratio is lower on the surface than in the inner areas due to slow extraction. In any case, the XPS analysis and the cyclic voltammograms at different Mg(NO_3_)_2_ concentrations reliably demonstrate the effective insertion-extraction of Mg^2+^ in the MgMn_2_O_4_ spinel structure. However, the electrode capacity decay in aqueous media evidences that, apart from the reversible insertion/extraction process, a concurrent reaction occurs, leading to structural changes that gradually accumulate over repeated cycles. These accumulated changes ultimately contribute to electrode degradation (see below).

Raman spectroscopy is a very useful tool for studying the insertion/extraction reactions in solid lattices, even when they are amorphous. This is particularly relevant in the case of the synthesized MgMn_2_O_4_ material, characterized by a small particle size with a limited crystallinity according to the XRD pattern. Furthermore, an electrochemical cell can be easily coupled to the Raman spectrometer (microscope) to acquire relevant operando structural information. The only caution is controlling laser beam power density on the sample to preserve its genuine structure [39].

Figure 4a shows the Raman spectra for the cathode in 1.0 M Mg(NO_3_)_2_ at different potentials: initially (at open circuit potential) and at −0.4, 0.0, 0.6, and 1.0 V. Before the electrochemical treatment, the electrode shows an ill-defined spectrum, which can be related with a low crystallinity degree, but also to the low Raman activity of MgMn_2_O_4_ as already shown in Figure 1b. The additional lack of definition in Figure 4a is due to other components in the electrode (binder and conducting carbon) and to the constraints of the spectroelectrochemical experimental configuration. As the electrode is polarized to positive potentials, four main bands are defined at 642, 575, 506, and 410 cm^−1^. The two high-wavenumber bands dominate the spectra, while the bands in the low-frequency region appear with a rather weak intensity. As previously mentioned, for the as-prepared MgMn_2_O_4_, the Raman band at 642 cm^−1^ can be assigned to the symmetric Mn-O stretching vibration of the MnO_6_ octahedra (A_1g_ symmetric mode). This peak shifts toward lower wavenumbers (from 652 to 642 cm^−1^) when the sample is oxidized, indicating a structural change due to Mg^2+^ extraction. In fact, in a simplified scenario, at positive potentials, the extraction of Mg^2+^ should lead to the oxidation of Mn(III) to Mn(IV) and, thus, to the generation of a MnO_2_-type solid according to Equation (2). The rest of the well-defined bands at 575, 506, and 410 cm^−1^ appearing at positive potentials are absent in the as-prepared MgMn_2_O_4_ sample, and they suggest the formation of a new phase, likely a birnessite layered structure because of the similarity with the Raman spectra reported for hydrated birnessite frameworks. The observed peaks are virtually coincident with those of MnO_1.86_·0.6H_2_O (506, 575 and 646 cm^−1^) [40].

Figure 4b shows the spectra evolution upon successive charge-discharge cycles. The morphology and the position of the peaks at the different applied potentials do not strongly depend on the scan number. This attests to the chemical reversibility of the processes occurring during Mg insertion/extraction. However, a closer inspection reveals that the original spectrum is not completely recovered; the spectrum of the reduced state becomes less defined, pointing to some changes in the structure with respect to that of the pristine material (Appendix A).

To obtain further information about the MgMn_2_O_4_ structural evolution, Mn 2p and O 1s XPS spectra were obtained for the charged (oxidized at 1.0 V) and discharged (reduced at −0.7 V) states after electrochemical activation (Figure 5). At first glance, the Mn 2p bands are shifted toward higher binding energies for the reduced and oxidized samples with respect to the native electrode (Figure 5a). This indicates an accumulation of Mn atoms with a higher oxidation state at the electrode surface. To obtain further information, the Mn 2p bands were deconvoluted by considering four main contributions (Appendix A) and the results are summarized in Table 2.

The XPS deconvoluted contributions at lower binding energies can be ascribed to Mn(III), while those at higher energies can be related to Mn(IV) [41]. The results show that Mn is present mainly as Mn(III) (83%) in the native electrode. The presence of 17% of Mn(IV) can be rationalized because the as-synthesized spinel is defective in Mg. The deconvolution indicates a significant increase in the content of surface Mn(IV) species, not only in the oxidized state but also in the reduced state. This points to the generation of MnO_2_-type solid domains electrochemically disconnected from the rest of the electrode. Such a change in the oxidation state is no longer observed after Ar^+^ bombardment (Figure 5b). In this case, a small new contribution appears at 647 eV, which can be related to the presence of Mn(II), probably generated during the etching.

The main changes in the XPS spectra are observed for the O 1s bands. Surface analysis (Figure 5c) indicates different oxygen species content as a function of the state of charge, while these differences nearly vanish after etching (Figure 5d). Table 3 summarizes the results obtained after peak deconvolution (Appendix A). Three main contributions corresponding to different oxygen species can be identified: Mn-O-Mn (structural oxygen) and Mn-OH, and H-O-H (oxygen coming from water) [42].

As observed, the Mn-OH contribution significantly increases in the reduced state, while the amount of H-O-H increases for both the reduced and the oxidized states. These results indicate the co-insertion of water molecules due to electrochemical activation. Note that a large amount of water is extracted during charging as the contribution of the Mn-OH band highly diminishes and nearly vanishes in the oxidized state. The co-insertion of water, which induces spinel hydroxylation, is reversible in agreement with the Raman results. On the other hand, the O 1s spectra for the etched samples show minor differences independently of the state of charge, as was the case of the Mn 2p band, indicating that not the whole electrode is being electroactive. This agrees with the fact that the XRD patterns are similar before and after electrochemical activation (Appendix A). This result contrasts with the electrode surface composition, which is drastically modified during the electrochemical treatment, as revealed by XPS.

Figure 6 shows FE-SEM images corresponding to a pristine MgMn_2_O_4_ film (Figure 6a,b), and after being submitted to either 17 (Figure 6c) or 120 (Figure 6d) charge-discharge cycles between −0.7 and 1.0 V in 1.0 M Mg(NO_3_)_2_. The native film comprises irregular micrometric aggregates, rapidly evolving toward a laminar structure. The size of the sheets increases with the number of electrochemical cycles. From the image analysis, an average width of 205 nm and an average thickness of 14 nm is estimated for the sheets after 17 scans (when the electrode shows the maximum electroactivity). In contrast, for cycle 120, the average sheet width doubles to 478 nm, with a thickness of 20 nm. Such a severe morphological change can be explained by a dissolution-redeposition process of Mn species (see below). In this context, it is essential to mention that elemental analysis of the electrolyte by ICP-OES at the end of the electrochemical experiments reveals the presence of dissolved manganese. These results agree well with previous studies on the insertion of Zn^2+^ in MnO_2_ cathodes, where the dissolution of manganese into the electrolyte via chemical disproportionation of Mn(III) species during the discharge process has been proposed [43]. Possible morphological changes as a function of the state of charge were studied by submitting two MgMn_2_O_4_ electrodes to 20 cycles of charge-discharge at 0.5 mA in 1.0 M Mg(NO_3_)_2_ and extracting one of them at −0.5 V and the other at 1.0 V. As observed, the apparent topography and the particle morphology for the oxidized and reduced samples are very similar (Figure 6e,f), which is expected as the reversible changes that occur in each cycle are unlikely to alter electrode topography and morphology on this scale significantly.

### 3.4. MgMn_2_O_4_ Modified with either Nickel or Cobalt

Transition-metal doping can improve electrochemical performance by altering cation ordering, suppressing the Jahn-Teller effect, and manipulating Mn^3+^ concentration. Modified MgMn_2_O_4_ electrodes were prepared using the same protocol but with the addition to the Pechini precursor solution of either Co(NO_3_)_2_·6H_2_O or Ni(NO_3_)_2_·6H_2_O. The modified electrodes, with a measured elemental composition corresponding to MgCo_0.71_Mn_1.25_O_4±δ_ and MgNi_0.73_Mn_1.34_O_4±δ_, are denoted for simplicity as MgCo_0.7_Mn_1.3_O_4_ and MgNi_0.7_Mn_1.3_O_4_, respectively. The XRD patterns reveal a low crystallinity degree and a phase change (Appendix A). The tetragonal structure of MgMn_2_O_4_ evolves towards an Fd-3m cubic structure [32,44]. On the other hand, regarding morphology, modified MgMn_2_O_4_ materials are characterized by quasi-spherical nanoparticles with an average size slightly smaller than that of pristine MgMn_2_O_4_: 7 nm for MgCo_0.7_Mn_1.3_O_4_ and 12 nm for MgNi_0.7_Mn_1.3_O_4_ (Appendix A).

The structural modification due to doping is expected to impact electrochemical performance significantly. As observed in Figure 7, the shape of the voltammograms and the potentials at which the Faradaic processes occur are similar for the three materials. Therefore, the manganese redox pair seems to be the most electrochemically active in the potential window studied: no clear peaks associated with the redox activity of cobalt or nickel species can be distinguished.

Figure 7d shows a comparison between the different materials for cycle 60. Current densities and, thus, gravimetric capacities obtained for the native MgMn_2_O_4_ spinel are larger than for the modified spinels. Interestingly, overpotentials are lower for the modified structures, with the Co-modified sample showing the highest reversibility. However, the CV definition is poorer than for the unmodified oxide.

An interesting aspect to be highlighted is the difference in the shape of the first CV for the different oxides. In the case of the MgMn_2_O_4_ spinel, an irreversible oxidation process (associated with activation) is observed starting from 0.7 V, while in the case of the modified samples, this oxidation signal is no longer observed. When nickel or cobalt atoms are introduced into the structure, a decrease in the Jahn-Teller effect is expected since the tetragonal phase partially changes to the cubic phase. The weakening in the Jahn-Teller effect leads to a decrease in the content of Mn(III) in the structure, favoring the presence of Mn(IV). Consequently, anodic activation is no longer apparent [45]. The presence of Mn(IV) in the modified samples can also be deduced from the appearance of reduction currents in the first negative-going scan, completely absent for the unmodified sample.

Table 4 quantitatively summarizes the main results for the pristine and modified spinels. Regarding cyclability, the structure with the greatest capacity retention is the spinel modified with nickel, MgNi_0.7_Mn_1.3_O_4_. In this case, at cycle 120, the capacity retention (with respect to the maximum value) is 77%, while the retention values for MgMn_2_O_4_ and MgCo_0.7_Mn_1.3_O_4_ are 49% and 60%, respectively. However, the capacity is lower for the modified spinels. This could be related to the smaller content of Mn, which is the active species as discussed previously. Although the modified samples are expected to be more stable, a change in the morphology toward a layered structure, similar to that seen upon cycling pristine MgMn_2_O_4_, is also observed (Appendix A). The results reported here agree with those previously published for Co-modified samples [32], showing that the introduction of Co or Ni atoms into the spinel structure does not improve the specific capacity of pristine MgMn_2_O_4_.

## 4. Discussion

As mentioned above, a major structural change occurs during the activation and subsequent cycling of MgMn_2_O_4_. Figure 8 shows the sketch proposed for explaining such a structure evolution. The fundamental unit in the crystal structure of MgMn_2_O_4_ polymorphs is the MnO_6_ octahedron. The units are linked via the edges and/or corners, generating a tunnel-like structure. Such a structure is expected to facilitate Mg^2+^ insertion/extraction during the electrochemical reactions. Raman results indicate that MgMn_2_O_4_ transforms into a birnessite-type structure characterized by layers of MnO_6_ octahedral units, as depicted in Figure 8. The initial extraction of Mg^2+^ induces a change in the manganese oxidation state from Mn(III) to Mn(IV) in agreement with XPS and Raman spectra. Such a reaction probably damages the structure stability, leading to the introduction of water molecules (and probably to nanoparticle disaggregation) according to:(3)$${\mathrm{MgMn}}_{2}O_{4}+{\mathrm{yH}}_{2}O\to {\mathrm{Mg}}_{1-x}\left({\mathrm{Mn}}^{3+}{}_{2-2x},{\mathrm{Mn}}^{4+}{}_{2x}\right)O_{4}\cdot {\mathrm{yH}}_{2}O+{\mathrm{xMg}}^{2+}+2{\mathrm{xe}}^{-}$$

Equation (3) is analogous to Equation (2), but it considers the introduction of water in the spinel framework.

In the subsequent negative-going scan, as the discharge reaction proceeds (i.e., reduction related to Mg^2+^ insertion), magnesium is incorporated into the partially deformed spinel structure resulting from the introduction of water during the charging process according to the Raman spectrum evolution and XPS results. During the intermediate stages of the discharge reaction, as more magnesium is inserted, the original structure tends to expand and form a layered phase (see Figure 8). Such an expansion should be favored by the presence of water molecules between the MnO_6_ octahedra layers and the coordination shell of the magnesium ions present in the electrolyte. Note that Mg^2+^ is identified as [Mg(H_2_O)_6_]^2+^ in aqueous solution [46]. The effective charge density of the inserted Mg^2+^ ions is diminished since in the complex [Mg(H_2_O)_6_]^2+^, the high charge of the central cation is partially screened by the coordinating H_2_O (or OH^−^) groups. This can promote Mg^2+^ mobility in the solid due to the diminution of its strong polarizing character in agreement with previous studies on the intercalation mechanism of zinc ions into MnO_2_ [47].

Further electrode reduction could eventually favor the insertion of additional water molecules. Upon reduction, the Mg^2+^ content is observed to double. It is also important to underline that no cathodic current is observed at potentials lower than 0.3 V for pristine MgMn_2_O_4_ electrodes during the first negative-going voltammetric scan. This result contrasts with a reduction reaction according to Equation (1), which should be discarded.

**Figure 8 materials-16-05402-f008:**
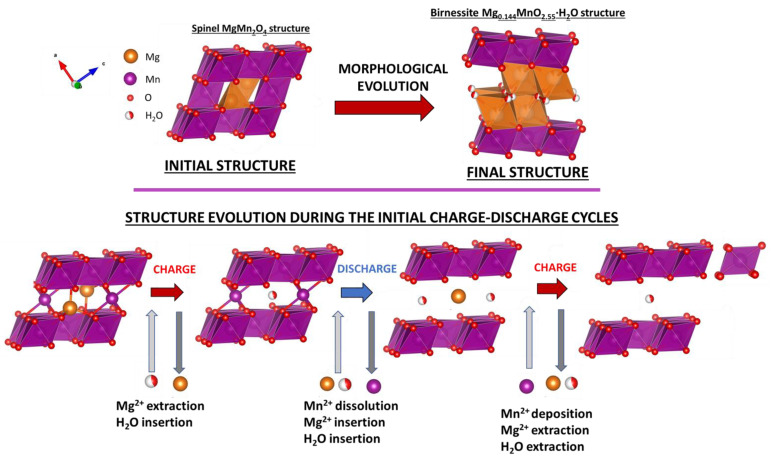
(**Top**): schematic illustration of the initial and final structure after repeated charge/discharge cycles. (**Bottom**): Simplified sketch depicting the MgMn_2_O_4_ spinel during initial charge-discharge cycling in aqueous Mg(NO_3_)_2_. Drawings were produced with VESTA-free software [48].

Together with magnesium insertion/extraction, other parallel reactions (apart from oxygen evolution) are likely favored by the complex electrochemical behavior of manganese oxides. Specifically, the reduction of MnO_2_ can also lead to the following:(4)$${\mathrm{MnO}}_{2}+H_{2}O+e^{-}\to \mathrm{MnOOH}+{\mathrm{OH}}^{-}$$
(5)$$\mathrm{MnOOH}+H_{2}O+e^{-}\to {\mathrm{Mn}}^{2+}+3{\mathrm{OH}}^{-}$$

These processes should be favored for the Mn atoms between the initial Mg^2+^ channels. This agrees with the electrode volume changes observed in the course of the in-situ Raman spectroscopy measurements with the confocal microscope (the focal distance needs to be readjusted to optimize the signal), indicating the dissolution of the electrode (Equations (4) and (5)). The dissolved Mn^2+^ can be reoxidized and deposited back to the electrode, generating the laminar structure observed by FE-SEM (see Figure 8). The similarity of the ex-situ X-ray diffractograms for the discharged and charged electrode indicates that this phase evolution starts from the surface, favored by the deposition of Mn^2+^ and that the electrode is not completely transformed.

In summary, the electrochemical reaction of the as-prepared MgMn_2_O_4_ cathode in aqueous 1.0 M Mg(NO_3_)_2_ proceeds via the structural transformation of the spinel phase to a hydrated birnessite-type layered structure. These phases, with different structures and multi-oxidation states, probably coexist during the complete Mg insertion/extraction process. This leads to tension inside the electrode and mechanical stress that damages its performance. On the other hand, excessive co-inserted water molecules could also lead to the collapse of the birnessite cathode structure.

Eliminating undesired phase transitions and freezing desired structures must be achieved to improve the spinel cathode performance. In this regard, doping with Co and Ni modifies the original MgMn_2_O_4_ tetragonal structure, but the modified oxides evolve during the electrochemical measurements in a similar way as the pristine spinel, probably due to insertion of hydrated Mg^2+^ and dissolution of Mn^2+^.

## 5. Conclusions

In this study, a spinel-type MgMn_2_O_4_ powder has been synthesized via a simple Pechini method. The electrochemical behavior studied in aqueous 1 M Mg(NO_3_)_2_ shows an oxidative activation process for magnesium insertion/extraction and a maximum capacity of 172 mAh g^−1^ measured at 0.5 mA (250 mA g^−1^). Combining in-situ Raman and ex-situ SEM images and XPS spectra gives insights into the formation of a layered structure due to the activation process, which involves incorporating water molecules and host hydroxylation. The layered structure presents a superior electrode performance due to its open structure and the charge-shielding effect of the inserted water molecules. However, the massive introduction of water in the structure and the mechanical stress due to the presence of different phases finally deteriorate the electrochemical performance. Doping with Co and Ni has proven to be a promising strategy for modifying the initial structure, resulting in enhanced capacity retention and cyclability, although insufficient for practical applications. The findings suggest the crucial role of water incorporation and structure hydroxylation in Mg^2+^ insertion/extraction. This highlights the need to explore the electrochemical behavior of these materials with other electrolyte solutions to reach both a deeper understanding of the underlying processes and progress toward Mg rechargeable batteries. It is worth noting that developing organic electrolytes for Mg batteries is an active area of research, and new formulations and combinations are being explored. Among the different organic electrolytes, one finds magnesium bis(trifluoromethane sulfonyl)imide (Mg(TFSI)_2_) in ethereal solvents (THF, DME), magnesium perchlorate (Mg(ClO_4_)_2_) in ethereal solvents (THF, DME), magnesium hexafluorophosphate (Mg(PF_6_)_2_) in carbonate-based solvents (ethylene carbonate or propylene carbonate) and magnesium trifluoromethanesulfonate (Mg(Tf)_2_) in glyme solvents among others. The Mn dissolution mechanism leading to the drastic structural change proposed in this work may be substantially hindered depending on the electrolyte nature. It is essential to extend the study presented here to other media. In this respect, the use of operando spectroelectrochemical methods is particularly important to monitor changes in the active material triggered by the electrochemical processes.

## Figures and Tables

**Figure 1 materials-16-05402-f001:**
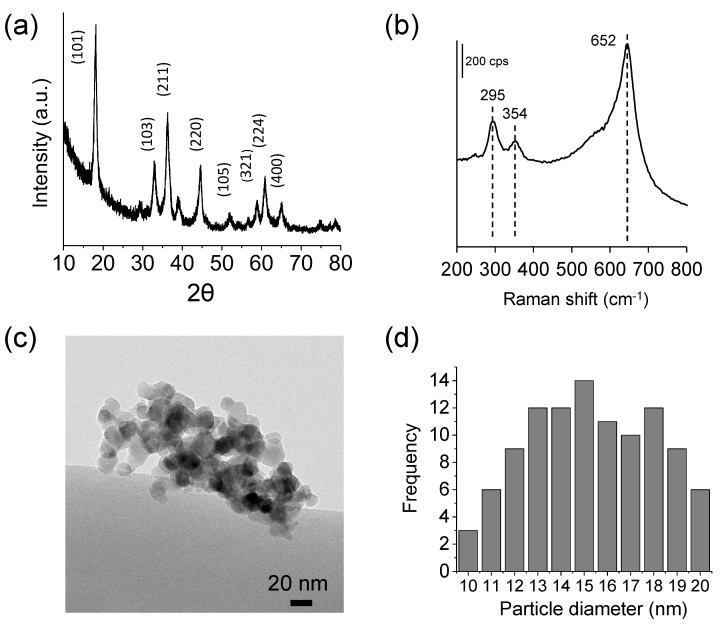
(**a**) XRD pattern, (**b**) Raman spectrum, (**c**) a representative TEM image for the as-prepared MgMn_2_O_4_ powder, and (**d**) a histogram for the particle size distribution.

**Figure 2 materials-16-05402-f002:**
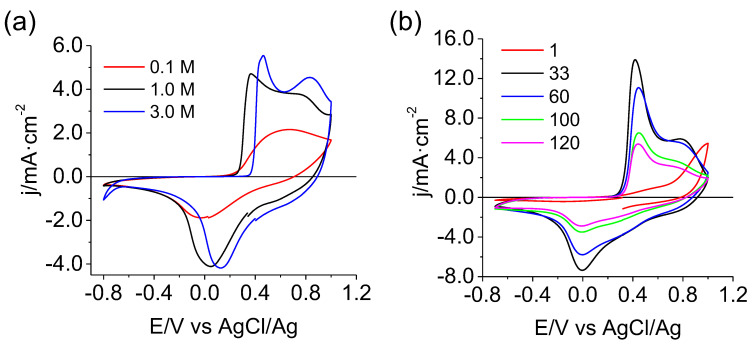
(**a**) CVs at 5 mV s^−1^ (cycle 10) for a MgMn_2_O_4_ electrode in contact with an aqueous electrolyte with different concentrations of Mg(NO_3_)_2_. (**b**) CV evolution with the scan number for 1.0 M Mg(NO_3_)_2_. For cycle 33, the spinel has the maximum capacity.

**Figure 3 materials-16-05402-f003:**
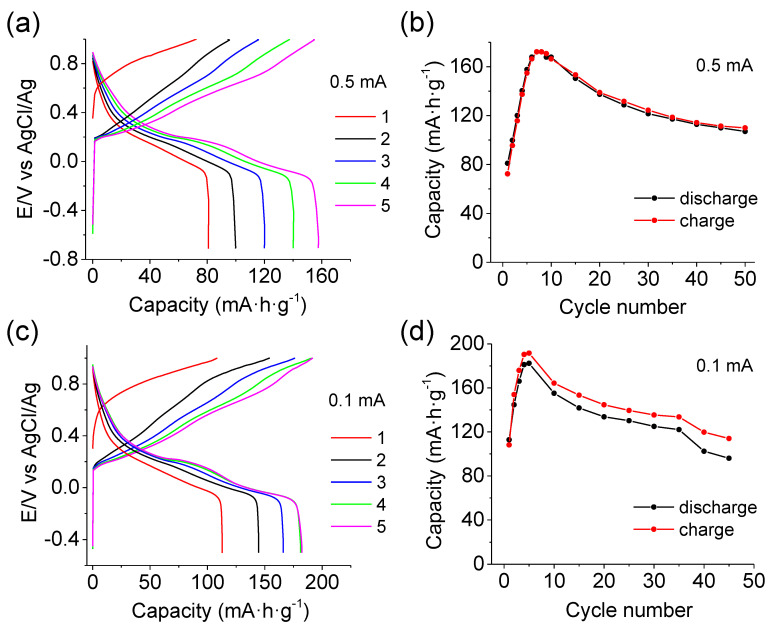
Charge-discharge cycles (**a**,**c**) with the corresponding cycling performance (**b**,**d**) for a MgMn_2_O_4_ electrode in contact with aqueous 1.0 M Mg(NO_3_)_2_ for two currents: 0.5 mA (250 mA g^−1^) (**a**,**b**) and 0.1 mA (53 mA g^−1^) (**c**,**d**).

**Figure 4 materials-16-05402-f004:**
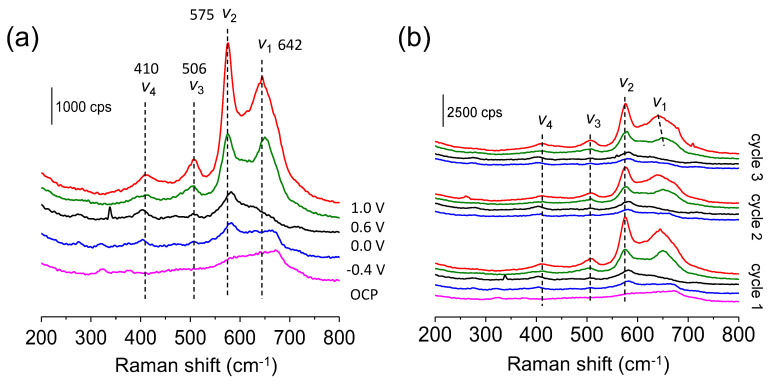
(**a**) Raman spectra for MgMn_2_O_4_ deposited on a glassy carbon electrode in contact with a 1.0 M Mg(NO_3_)_2_ electrolyte at five different applied potentials: −0.4 V (blue), 0.0 V (black), 0.6 V (green), 1.0 V (red) and at open circuit potential (OCP, pink). (**b**) Raman spectra evolution with the cycle number for different applied potentials.

**Figure 5 materials-16-05402-f005:**
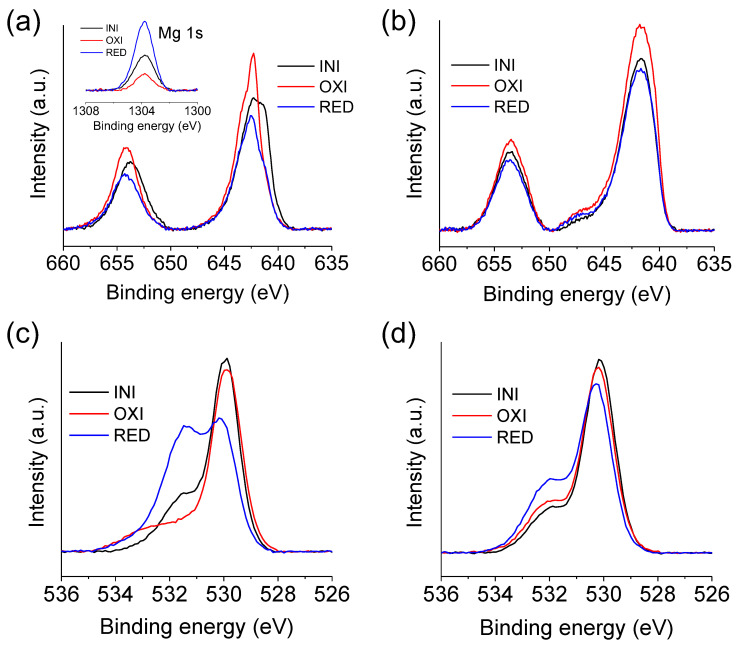
Mn 2p (**a**,**b**) and O 1s (**c**,**d**) XPS for MgMn_2_O_4_ at different states of charge: reduced at −0.7 V (RED) and oxidized at 1.0 V (OXI) in 1.0 M Mg(NO_3_)_2_ after 15 CVs at 5 mV s^−1^. The spectrum for a pristine MgMn_2_O_4_ sample (INI) is also included for comparison. Spectra are shown for surface (**a**,**c**) and 20 nm in-depth analysis (**b**,**d**). The inset in panel (**a**) shows the Mg 1s spectra for the different states of charge.

**Figure 6 materials-16-05402-f006:**
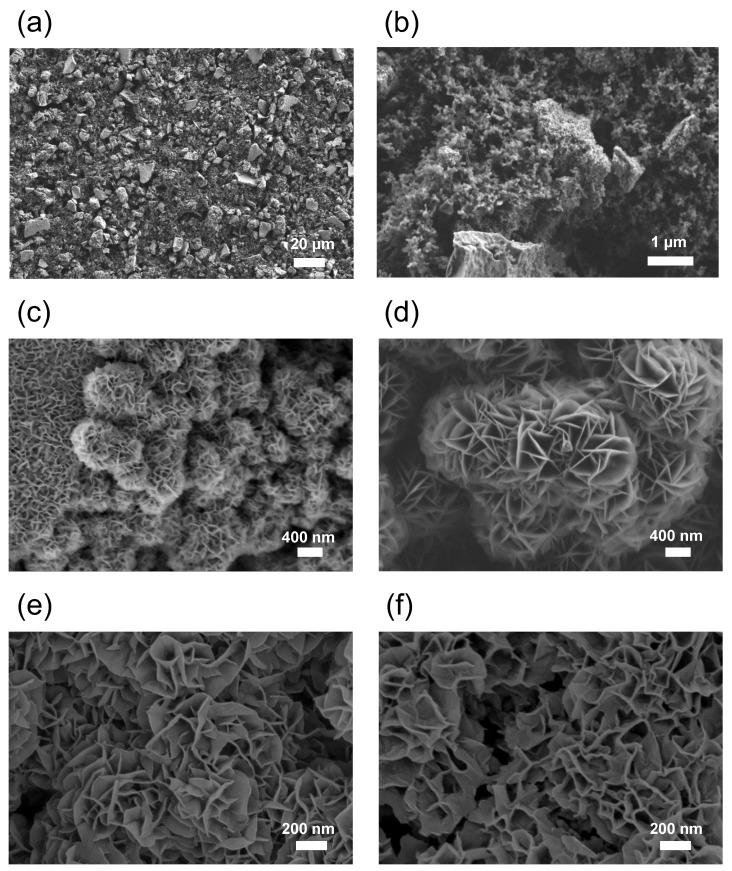
FE-SEM images for the native film before electrochemical cycling (**a**,**b**) and after 17 (**c**) and 120 cycles (**d**). Images for electrodes extracted in the reduced state at −0.5 V (**e**) and in the oxidized state at 1.0 V (**f**) after 20 cycles of charge-discharge at 0.5 mA are also included.

**Figure 7 materials-16-05402-f007:**
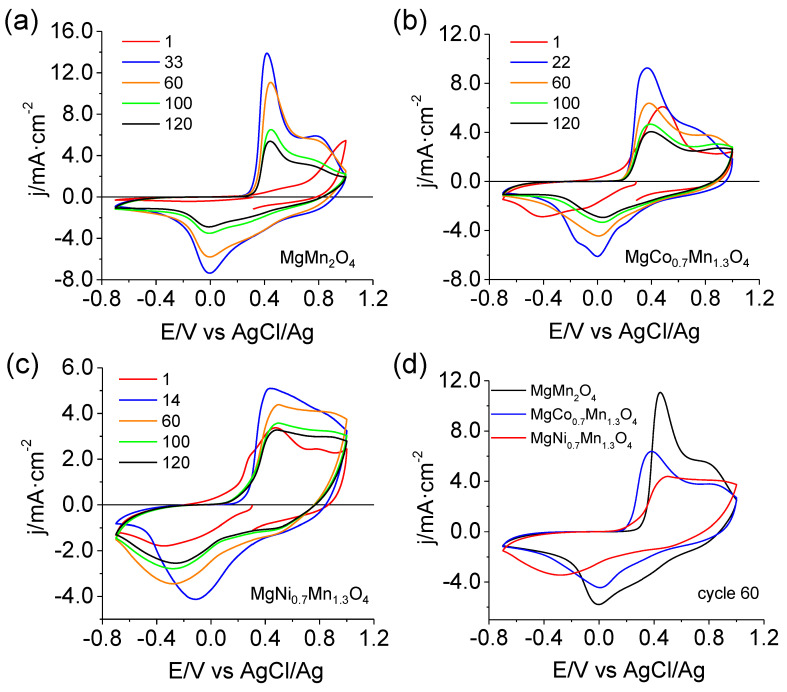
CVs at 5 mV s^−1^ in 1.0 M Mg(NO_3_)_2_ for pristine MgMn_2_O_4_ (**a**), MgCo_0.7_Mn_1.3_O_4_ (**b**), and MgNi_0.7_Mn_1.3_O_4_ (**c**). The 60th CVs for the three oxides are also shown (**d**) for comparison.

**Table 1 materials-16-05402-t001:** Mg/Mn ratio calculated from the XPS peaks for the Mg 1s and Mn 2p transitions for a sample either pristine or extracted from a 1.0 M Mg(NO_3_)_2_ aqueous solution at either 1.0 V (oxidized) or −0.7 V (reduced). The table also shows the results obtained after etching the samples (about 20 nm by Ar^+^ bombardment). (The corresponding spectra are shown below).

	Ratio Mg/Mn Surface	Ratio Mg/Mn after Etching
Pristine MgMn_2_O_4_	0.42	0.54
Oxidized MgMn_2_O_4_	0.16	0.27
Reduced MgMn_2_O_4_	0.96	0.78

**Table 2 materials-16-05402-t002:** XPS analysis of Mn 2p for the as-prepared electrode and after electrochemical activation by cyclic voltammetry in 1.0 M Mg(NO_3_)_2_ and extraction at either 1.0 V (oxidized) or −0.7 V (reduced).

		Mn(III)		Mn(IV)	
Pristine MgMn_2_O_4_	Mn 2p peak (eV)	641.1	642.5	644.4	646.5
	Content (%)	83		17	
Oxidized MgMn_2_O_4_	Mn 2p peak (eV)	641.2	642.2	643.2	644.9
	Content (%)	44		56	
Reduced MgMn_2_O_4_	Mn 2p peak (eV)	641.4	642.4	643.5	644.8
	Content (%)	56		44	

**Table 3 materials-16-05402-t003:** XPS analysis of the O 1s signal for the as-prepared sample and after CV electrochemical activation in 1.0 M Mg(NO_3_)_2_ and extraction at either 1.0 V (oxidized) or −0.7 V(reduced).

		Mn-O-Mn	Mn-OH	H-O-H
Pristine MgMn_2_O_4_	O 1s peak (eV)	529.9	531.5	532.9
	Content (%)	70	28	2
Oxidized MgMn_2_O_4_	O 1s peak (eV)	529.9	531.4	533.0
	Content (%)	73	16	11
Reduced MgMn_2_O_4_	O 1s peak (eV)	530.0	531.4	532.7
	Content (%)	40	50	10

**Table 4 materials-16-05402-t004:** Capacity and capacity retention values after 120 cycles for each spinel.

	Max. Capacity (mAh g^−1^)	Capacity for Scan 120 (mAh g^−1^)	Retention Capacity for Scan 120 (%)
MgMn_2_O_4_	144	71	49
MgCo_0.7_Mn_1.3_O_4_	118	71	60
MgNi_0.7_Mn_1.3_O_4_	94	72	77

## Data Availability

The data will be made available on request from the corresponding authors.

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
