# Peer review of "Unraveling the Phase Transition Behavior of MgMn2O4 Electrodes for Their Use in Rechargeable Magnesium Batteries"

_materials, 2023, doi:10.3390/ma16155402_

Round 1

Reviewer 1 Report

The introduction to the authors is well written.  At the end of the introduction, you write "However, it is highly relevant to progress in the development of magnesium batteries." It would be good to expand on this and write why research on this issue is important and what it will bring.

In the methodology section, more attention should be paid to characterization methods. Expand section 2.2 and indicate in what modes the equipment for studying the structure worked.

You have studied the magnesium insertion-disinsertion from MgMn2O4 in an aqueous medium. But why didn't you list possible other environments? Why not give a short comparison with other environments.

Figure 2 could indicate reduction cycles and oxidation cycles.

On lines 292-294 you write about reversible changes. But at the same time, write that the structure of the particles is very similar, but apparently not completely identical. Then the changes will be reversible but not completely. Here we can talk about a large number of charge-discharge cycles. But still, changes will accumulate over time.

At the end of section 4, you should add information about the practical application of the results of the work. Compare the results obtained with other types of batteries and with the results of other researchers.

The same is true for conclusions. Add information about practical implementation.

Reviewer 2 Report

The manuscript entitled “Unraveling the phase transition behavior of MgMn2O4 electrodes for their use in rechargeable magnesium batteries” employed in-situ Raman spectroscopy and other ex-situ physicochemical characterization techniques to investigate the Mg2+ insersion and extraction mechanism during charge and discharge process. They have found that a progressive structure evolution from a well-defined spinel to a birnessite-type arrangement takes place during the first cycles, accompained by obvious morphological change. The impact of Ni and Co doping on the electrochemical performance of MgMn2O4 electrodes is also discussed accordingly. The manuscript warrants the journal’s aim and scope. All the experimental data are accuritely displayed in the text with proper disscussions. I recommend MAJOR REVISION before it can be accepted for publication. The following comments are listed for authors’ references.

1. Please remove the background of the XRD spectra in Figure 1a and Figure S5. Also, please provide the JCPDS number of the phase used for indexing the patterns.

2. In Figure 1b, the authors compared the vibration bands of LiMn2O4 and MgMn2O4. Correspondingly, the Raman spectra of LiMn2O4 should be provided in the same coordinate system.

3. The authors have mentioned that the lack of stoichiometry for MgMn2O4 prepared through sol-gel method. How does this lack of stoichiometry affect the insertion/extraction of Mg2+ cations?

4. Why the columbic efficiency of the battery is even lower under small current density?

5. Table 1 listed the relative ratio of Mg/Mn elements in MgMn2O4 before and after etching process through XPS analysis. Please provide the related XPS spectra used for such calculation.

6. In the XPS spectra in Figure 5, the authors mentioned that the Mn 2p bands are shifted toward higher binding energies for both the reduced and oxidized states compared to the native electrode. This is quite uncommon in the XPS spectra because the shift of peaks normally corresponds to the change of oxidation states. The valence state of Mn will increase and decrease during the oxidation and reduction of the electrode material, why do the peaks all move towards high binding energy direction?

7. In Figure S.2, every spin-splitting peak have been fitted to 4 individual peaks with green and red color. What do the four peaks stand for in the spectra?

8. Please provide the integral area of each fitted peak in Figure S.2.

9. Why the XPS spectra of MgMn2O4 reduced at -0.7 V in 1.0 M Mg (NO3)2 after 15 CVs reveal quite different characteristics? Why not use the XPS spectra of etched surface to illustrate the change of O 1s spectra during oxidation and reduction?

10. What are the differences for the morphologies in Figure 6 c, d and e, f? Why there are siginificant morphology change for the MgMn2O4 electrodes? Only because of the dissolution-redeposition of Mn?

11. If the redox peaks in CV curves are higher for MgMn2O4 electrodes than the modified samples, what is the significance of doping modification?

12. The authors finally discussed the insersion and extraction mechanism of Mg2+ in MgMn2O4 electrodes. What is the impact of Co and Ni doping on the insersion and extraction of Mg2+?

Some professional phrases should be explaned more accurately.

Reviewer 3 Report

The work is dedicated to research on MgMn2O4 cathodes for Mg-ion batteries. Such studies have been carried out by other authors, but there is undoubtedly still much to be done and the topic deserves attention. The authors point to the problems encountered when using MgMn2O4 electrodes in aqueous solutions, the challenges in ensuring high capacity, and poor cyclability. The authors have mitigated some problems by doping with Co and Ni, but, as they openly admit, the performance of the MgMn2O4 electrodes is not satisfactory. The authors have not clearly articulated the contribution their work makes to the development of MgMn2O4 electrodes, nor what goals they set themselves in undertaking the research described in the manuscript.

Comments on the text are presented below.

1)      2.3 Electrochemical experiments

Information on the manufacturer of carbon black Super P, and PVDF is missing. Is PVDF an abbreviation for polyvinylidene fluoride? Information about the size of the electrode is also lacking. What form did the working electrode take? Was it a layer of MgMn2O4 nanoparticles-carbon black Super P-PVDF mixture deposited on a titanium substrate? If so, what was its thickness? For the convenience of the reader, the authors could add a schematic of such an electrode.

2)      3.2. Electrochemical behaviour of MgMn2O4 in an aqueous electrolyte of Mg(NO3)2

‘After 5 cycles, two oxidation peaks and one broad reduction peak are defined containing different contributions, while currents increase gradually upon cycling’ - This sentence is unclear, please write it in a different way.

3)      Fig. 8. Is the reference from the literature [50] correct?

4)      What criterion did the authors follow in presenting the cyclic curves? Cyclic curves recorded as the tenth, sixty-ninth, etc., have been presented.

5)      The description of Fig. S.4 is incorrect; there is no curve labelled c. The composition of the solution in which the electrodes were immersed and electrochemical tests were conducted is missing. There is no comment on why the titanium peaks are visible in the figure. Information on titanium support is provided only in the ‘Materials and Methods’ section. Why is titanium present in the material tested by XRD? The diffraction pattern labelled Ti differs from the values for titanium known from JCPDS card No. 44-1294 (https://link.springer.com/article/10.1007/s11663-000-0110-3#Abs1), what could explain this difference? In the diffractograms recorded after charging and discharging, there are no characteristic peaks for MgMn2O4. What could be the reasons for this?

6)      For the reader's convenience, the formulas of the studied compounds could be added next to the curves in the panel d of Figure S5.

A moderate revision of the English language usage is required. A problematic sentence was addressed in the review. Consider using a term other than "thermally treated in a stove," as a stove is typically a household appliance, not a laboratory oven.

Round 2

Reviewer 1 Report

The authors finalized the article and answered my questions. It would be good to move the answer to question 5 in a little more detail to the text of the article.

Author Response

Thanks for the comment. 

In the revised version, we have added a couple of additional sentences in section 3.3 as suggested by the reviewer (see p. 8-9). However, we have refrained from adding a more comprehensive discussion as it is included later in the MS. On p. 8-9, it now reads:

“… of Mg2+ in the MgMn2O4 spinel structure. However, the electrode capacity decay in aqueous media evidences that, apart from the reversible insertion/extraction process, a concurrent reaction occurs, leading to structural changes that gradually accumulate over repeated cycles. These accumulated changes ultimately contribute to electrode degradation (see below).”

Reviewer 2 Report

In the current version, the author has revised some of the issues raised in the previous draft, but there are still some related issues that affect readers' understanding in the manuscript that need further revision, which are listed for authors’ references:

1. All the curves in the manuscript lack vertical coordinates and should be supplemented.

2. The Raman spectra of LiMn2O4 used in this work should be provided to compare with that of MgMn2O4 to illustrate the difference of vibration bands and the related effects on the electrochemical performance.

The manuscript can be accepted for publication after the above addressed issues have been modified.

Author Response

  1. All the curves in the manuscript lack vertical coordinates and should be supplemented.

We have modified the figures. We have added a vertical axis for the XPS and XRD patterns. Additionally, for the Raman spectra, we have included a scale bar.

In the case of XPS and XRD, we have only specified that the y-axis corresponds to the intensity in arbitrary units. Different factors such as detector efficiency, sample thickness, sample location, etc. can lead to slight intensity differences, even though we have consistently used the same experimental setup and conditions for all samples.

Regarding the Raman spectra, we have introduced a scale bar indicating the counts per second (cps), as the spectra are acquired sequentially for the same unique sample.

  1. The Raman spectra of LiMn2O4 used in this work should be provided to compare with that of MgMn2O4 to illustrate the difference of vibration bands and the related effects on the electrochemical performance.

In the revised version of the MS, the LiMn2O4 Raman spectrum has been introduced in the “Supplementary Materials” (Figure S1). We have used references 36 and 37 to interpret and assign the Raman bands for MgMn2O4.

Comparing the electrochemical behaviors of MgMn2O4 and LiMn2O4 is not straightforward as LiMn2O4 has been investigated in organic electrolytes, whereas in our case, aqueous media are employed, and water molecules appear to play a crucial role in the charge-discharge process.

Reviewer 3 Report

The authors have addressed my comments in a satisfactory way. However, Figure S4 still raises my doubts. Could the authors add an XRD interpretation to Figure S4, as they did in response to point 5 of my review? If the authors used Origin software to construct figures, they could use a free Origin app (XRD analysis) that can help to prepare such XRD plots with interpretation, similar to that provided by XRD instruments.

Author Response

We thank the reviewer for his/her comment. In the revised version of the Supplementary Materials, Figure S4 has been modified based on the reviewer’s comment. In the caption, reference is now made to the JCPDS Cards employed for the identification of both Ti and MgMn2O4. In addition, we have also included two additional panels in which the contributions of Ti has been subtracted (panels c) and d), corresponding to panels a) and b), respectively).